# Effect of Heating Oxidation on the Surface/Interface Properties and Floatability of Anthracite Coal

**Guoqiang Rong [1,2], Mengdi Xu [1,2,*], Dongyue Wang [3], Xiahui Gui [1,2,*] and Yaowen Xing [1,2,*]**

[1]  Chinese National Engineering Research Center of Coal Preparation and Purification,
    China University of Mining and Technology, Xuzhou 221116, China; cumtrgq@126.com
[2]  School of Chemical Engineering and Technology, China University of Mining and Technology,
    Xuzhou 221116, China
[3]  School of Mineral Processing and Bioengineering, Central South University, Changsha 41000, China;
    dywang@csu.edu.cn
*   Correspondence: cumtxmd@outlook.com (M.X.); guixiahui1985@163.com (X.G.); cumtxyw@cumt.edu.cn (Y.X.)

**Abstract:** Oxidation processes of coal surfaces are both fundamental and interesting from academic and engineering points of view. In this work, we comprehensively analyzed the mechanism of heating oxidation at 200 °C on the surface/interface characters and the floatability of anthracite coal. The variations of surface/interface characters were studied using SEM (scanning electron microscopy), FTIR (Fourier transform infrared spectroscopy), and XPS (X-ray photoelectron spectroscopy). The floatability was further identified using Induction Time and Bubble-Particle Wrap Angle. It was found that, after heating oxidation at 200 °C, both surface ravines and oxygen-containing groups were increased. The degradation of hydroxyl on anthracite could be neglected during the heating, while the oxidation of hydrocarbon chains dominated the balance of hydrophobicity and hydrophilicity on coal surface. The induction time significantly increased from 200 ms to 1200 ms and 2000 ms after 10 h and 20 h of heating oxidation at 200 °C, respectively. Additionally, raw coal exhibited the fastest kinetics of bubble-particle attachment and the largest wrap angle, directly proving that the floatability decreased after oxidation.

**Keywords:** heating oxidation; surface/interface properties; floatability; induction time; bubble-particle wrap angle

## 1. Introduction

In China, the coal is currently the important energy source. Flotation is one of the most common methods for fine coal de-ashing that cannot be achieved by gravity separation [1–4]. It is based on different wetting properties between organic matters and gangue particles. Coal is inherently hydrophobic. Therefore, it can be easily captured by air bubbles in the water phase. Conversely, gangue particles such as silica and clays are inherently hydrophilic. As a result, they cannot get attached onto a bubble surface, sinking to the bottom of the flotation cell to become the tailings [3].

Oxidation processes of coal surfaces are both fundamental and interesting from academic and engineering point of view [5]. During the storage of coal slime, coal can be oxidized easily due to weather processes. On the other hand, coal spontaneous combustion can also make the coal particles oxidized. A great deal of heat is released during spontaneous combustion, leading to a serious oxidation of coal particles. Coal floatability and recovery decreased significantly after oxidation and cannot even be economically recovered via a traditional flotation method [6–11]. It is well known that oily collectors have been always used for coal flotation. However, the adsorption efficiency of oil collectors on oxidized coal surfaces was decreased significantly. To float oxidized coals, a number of works on polar collector, pretreatment, and new flotation procedures have been reported [6,12–16].

To understand the formation process of oxidized coal, the effect of high-temperature oxidation by heating (>500 °C) on the surface/interface characters and wettability of high-ranked coal and anthracite coal has been widely investigated [11,17]. It was found that the hydrophobicity always decreases after oxidation by heating since the content of hydrophobic functional groups decreases, while the number of hydrophilic groups increases [11,17]. One opposite finding is that, for low-ranked coal, the contact angle increased after oxidation by heating after decreasing in hydrophilic functional groups [15,18,19]. It is well known that two opposing phenomena occur that influence the hydrophobicity at the coal surface during oxidation processes with heat treatment. One is the degradation of oxygen-containing groups such as hydroxyl and carboxyl, thereby increasing the hydrophobicity. The other is the oxidation of hydrocarbon chains on the coal surface, producing new carboxyl and phenol groups. This in turn makes coal more hydrophilic [18,20]. Çınar [18] found that the improved floatability of low-ranked coal after oxidation by heating was mainly attributed to the removal of OH groups. It is speculated that heating for too long could oxidize low-ranked coal, hence decreasing its floatability. An open question is whether the hydrophobicity of high-ranked coal such as anthracite could also be enhanced by oxidation by heating. In previous works, it is noted that the heating temperature for treating anthracite was very high (>500 °C). Heating oxidation at 200 °C for anthracite coal has not been reported.

The objective of this study is to comprehensively understand the effect of oxidation on the surface/interface characters and floatability of anthracite by heating (200 °C). The changes of surface/interface characters of coal particles were confirmed using SEM, FTIR and XPS. The floatability of coal particles was further identified using induction time and the bubble-particle wrap angle.

## 2. Materials and Methods

### 2.1. Materials and Heating Oxidation Process

The anthracite sample was obtained from Xuehu Plant, China. The raw samples were crushed into small particles by a hammer mill, whose particle size was smaller than 0.5 mm. The ash content of anthracite was 21.30%. A muffle furnace was used for heating the coal sample. The interval of heating was 10 h and 20 h at 200 °C, whose process was the heating oxidation, respectively.

### 2.2. SEM

An SEM system was used to study the changes on surface morphology and elemental composition of coal before and after heating oxidation at 200 °C. The type of SEM system was FEI Quanta TM 250 (Hillsboro, OR, USA) which was equipped with Energy Dispersive X-ray Spectrometer (EDS). Gold sputtering treatment was carried out before the SEM test.

### 2.3. FTIR

FTIR (Nicolet is5, Thermo Scientific, USA) tests were performed in a wave number range of 4000 $cm^{-1}$ to 400 $cm^{-1}$ before and after heating oxidation at 200 °C. A coal sample of 2 mg mixed with KBr of 300 mg was ground into −2 μm. The pressure that was applied to prepare a thin circular plate was 30 MPa.

### 2.4. XPS

XPS system with 900 μm light spot size and Al Ka radiation (hv = 1486.6 eV) were used to examine the surface chemistry changes before and after heating oxidation at 200 °C. The type of XPS system was ESCALAB 250Xi, Thermo Scientific, Waltham, MA, America. XPS Peakfit software was used for data analysis [21]. The binding energies were calibrated with respect to the C1s hydrocarbon (-CH$_2$-CH$_2$-bonds) peak set to 284.6 eV.

### 2.5. Induction Time Test

A homemade induction time measuring system [21,22] was employed for determining the induction time between bubble and anthracite coal particles. This system contains a controller, high-speed camera, micro-syringe, micro-displacement sensor, actuator, illumination device, measuring cell, etc., as shown in Figure 1. The coal, whose particle size was 0.5–0.25 mm, was sieved for preparing a flat particle bed. A bubble produced at the end of capillary tube was driven to contact with the particle bed and then kept in contact for a prescribed time. The induction time was defined as the minimum contact time at which the attachment happened.

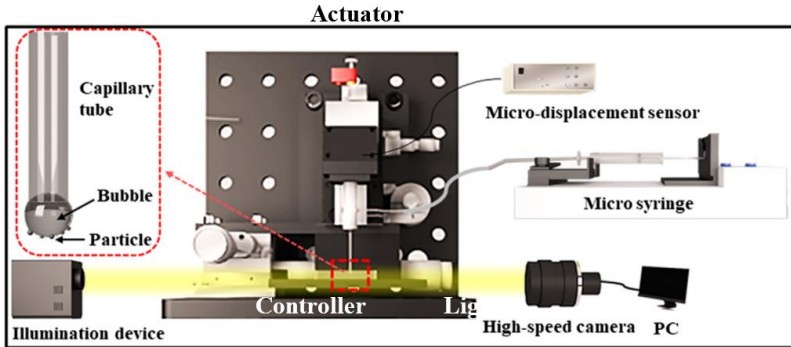

**Figure 1.** The induction time measuring system ([22], Reproduced with permission from Xia Y., Fuel; published by Elsevier, 2019).

### 2.6. Bubble-Particle Wrap Angle Experiment

The bubble-particle wrap angle was used to reveal coal floatability before and after heating oxidation at 200 °C. A schematic of the procedure to measure the Bubble-particle wrap angle ($\theta$) is shown in Figure 2. In order to simulate the flotation process and indicate the situation of particles who attached onto bubble surfaces, the bubble-particle wrap angle experiment was introduced by Chu et al. firstly [23]. It provides the average coverage angle of the coal particles on bubble. The bubble-particle wrap angle (BPWA) was directly connected to the floatability of the coal particles. Attachment kinetics showed that better floatability occurred following the larger BPWA. 2 g of coal (particle size was 0.25–0.125 mm) was put into a $5 \times 5 \times 10$ cubic cm container, together with 200 mL of ultrapure water. A rotor speed of 250 rpm was set for the magnetic stirrer during the experiment. The stirring time with rotatory period was set as 20, 40, 80 and 160 s, and bubble images were recorded accordingly. MB-Ruler software was used to analysis and measure the wrap angle.

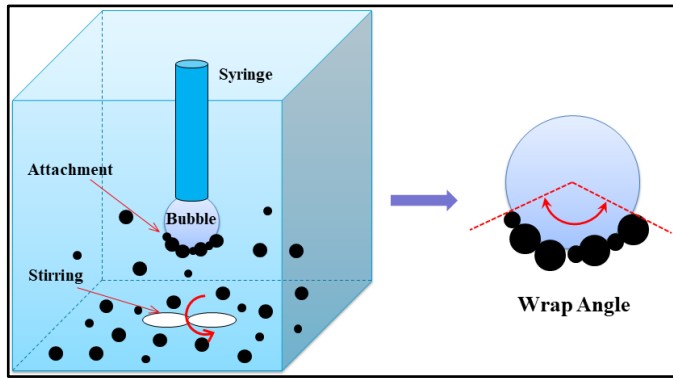

**Figure 2.** Schematic of bubble-particle wrap angle experiment ([24], Reproduced with permission from Xia Y., Fuel; published by Elsevier, 2019).

## 3. Results and Discussion

### *3.1. Effect of Heating Oxidation at 200 °C on Surface Morphology and Functional Groups of Anthracite Coal*

The surface morphologies of anthracite coal before and after heating oxidation at 200 °C are shown in Figure 3. The EDS plane sweep and the chemical compositions before and after heating oxidation at 200 °C from the EDS plane sweep are depicted in Figures 4 and 5. LH-10 and LH-20 represented the coal sample after heating oxidation for 10 h and 20 h, respectively. As shown in Figure 3, the raw coal surface was quite smooth, while the surface after heating was characterized by lots of ravines and asperities. According to the Wenzel model, the hydrophilicity can be enhanced by surface roughness on the hydrophilic surface. The contact angle of the raw coal was approximately 80–90°, showing a relative high hydrophobicity, while the increased surface roughness was detrimental to the floatability. From EDS results, the content of organic carbon decreased from 74.58% to 51.32% and 46.01% after 10 h and 20 h of heating, respectively. In contrast, the oxygen content increased from 17.65% to 35.55% and 38.07%. This proves that the coal surface was significantly oxidized during the heating process. The organic matters were attacked by oxygen, generating oxygen-containing groups such as hydroxyl, carbonyl, and carboxyl. These oxygen-containing groups further decomposed into gases such as carbon dioxide, leaving inorganic minerals on the surface of anthracite coal. As a result, a lot of ravines were formed.

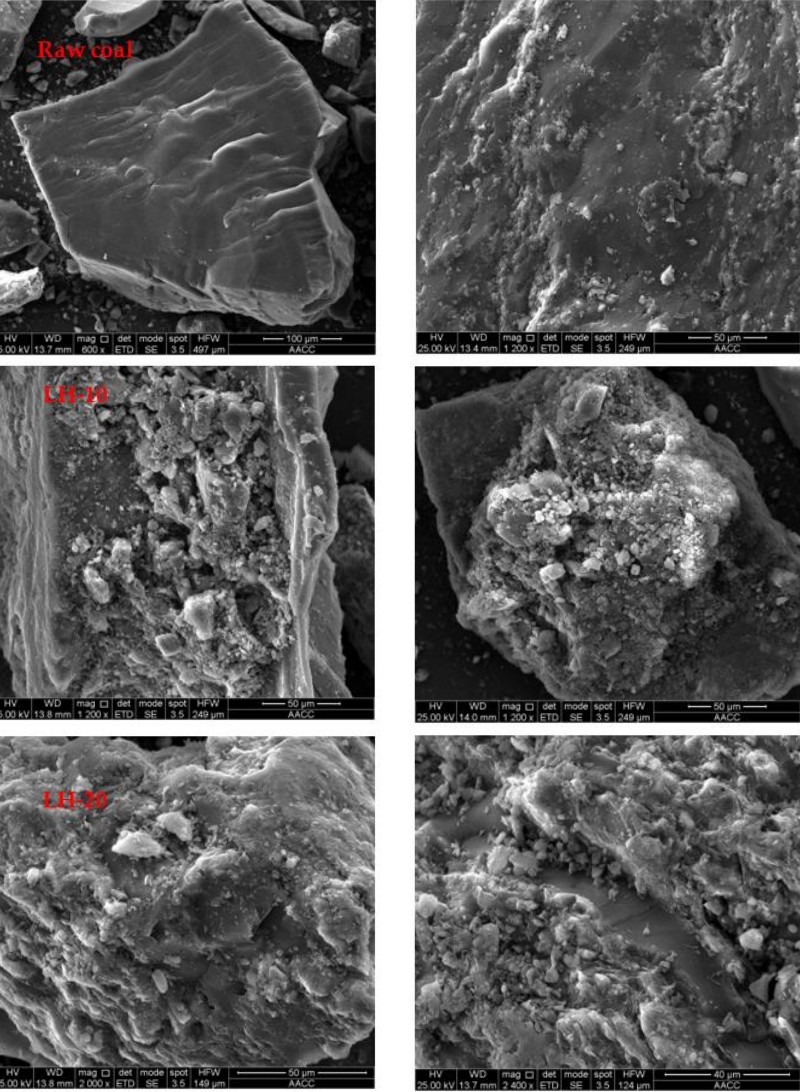

**Figure 3.** Surface morphologies of coal before and after heating oxidation at 200 °C.

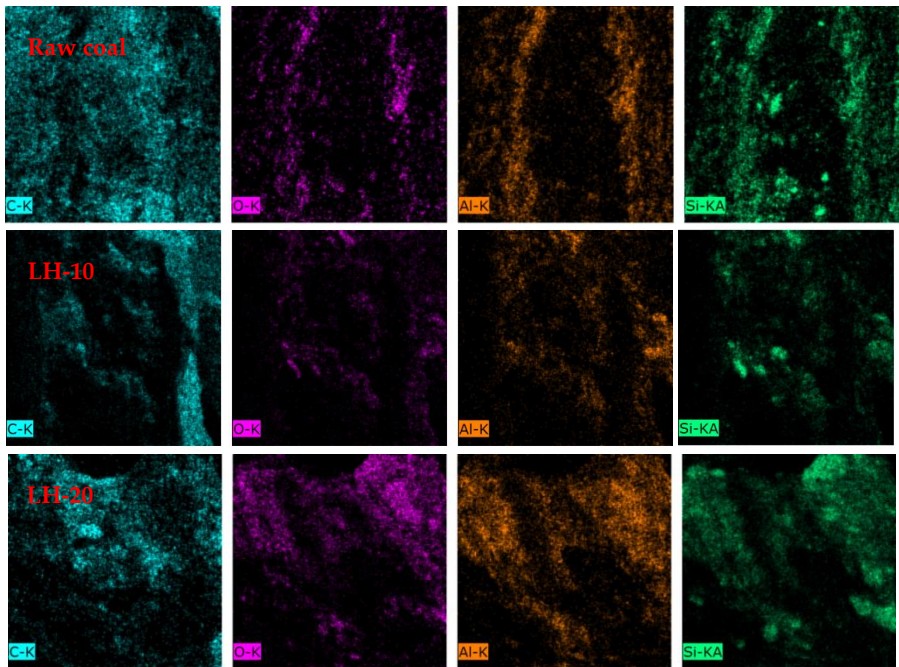

**Figure 4.** EDS plane sweep of the coal sample before and after heating oxidation at 200 °C.

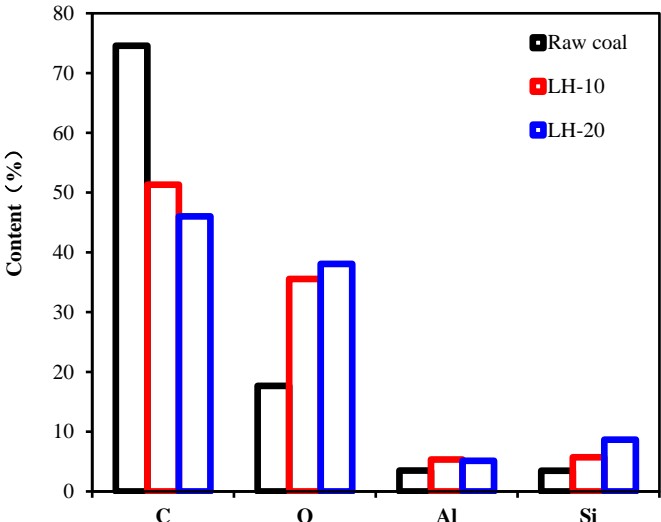

**Figure 5.** Chemical compositions before and after heating oxidation at 200 °C from EDS plane sweep.

FTIR of the coal before and after heating oxidation at 200 °C are shown in Figure 6. The peak at 3440 cm$^{-1}$ can be attributed to the –OH group, whereas the peak at 1700 cm$^{-1}$ can be attributed to the C=O group. The peaks at 3440 cm$^{-1}$ and 1700 cm$^{-1}$ both enhanced after heating. This means that the number of hydrophilic oxygen-containing groups increased, which is consistent with EDS results. Other research also indicated that the oxidization will lead to the increase in the hydrophilic oxygen-containing groups by using FTIR [25,26]. Unlike low-ranked coal, the number of –OH and C=O groups on anthracite surface did not decrease after heating, mainly because a small amount of these groups pre-existed on raw anthracite surface. Therefore, the degradation of hydroxyl could be neglected while the oxidation of hydrocarbon chains dominated the balance of hydrophobicity and hydrophilicity on an anthracite coal surface.

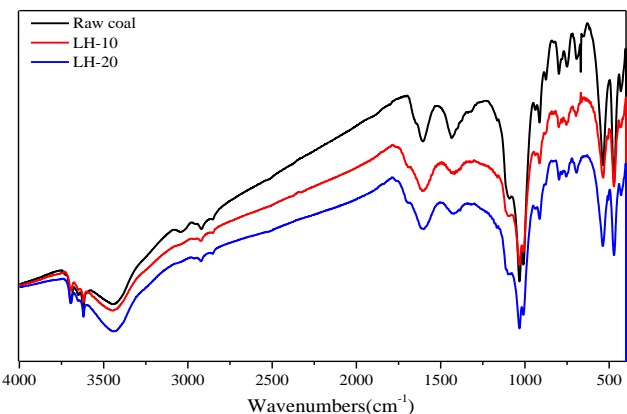

**Figure 6.** FTIR of the coal before and after heating oxidation at 200 °C.

XPS spectrums of the coal before and after heating oxidation at 200 °C are shown in Figure 7. The semi-quantitative results of the surface chemical composition from XPS wide spectrums show that the content of organic carbon decreased from 45.68% to 34.11% and to 28% after heating intervals of 10 h and 20 h, respectively. In contrast, the oxygen content increased from 36.54% to 44.27% and to 48.69%. Peak fitting of the XPS C1s peak provided quantitative results about the functional groups on the coal surface, as shown in Figure 8. The binding energies of O=C-O and C=O, C-O, C-H, and C-C were 289.1 eV, 286.6 eV, 285.60 eV, and 284.60 eV, respectively [23]. Functional groups of the surface of coal before and after heating oxidation at 200 °C from peak fitting of the XPS C1s peak are shown in Figure 9. The hydrophobic functional group C-C/C-H decreased after heating, while the total content of hydrophilic oxygen-containing groups (C-O, C=O, and COOH) increased. Remarkably, the content of COOH increased from 3.49% to 6.65% and to 6.67% after heating intervals of 10 h and 20 h, respectively, which is consistent with FTIR results. Xia et al. [11] also found that the heating process reduces the content of hydrophobic functional groups (C-H and C-C) but increases the content of hydrophilic functional groups (C-O, C=O and COOH). This change in the chemical groups led to the decrease in the hydrophobicity, which is closely related to the coal flotation. Based on the above analysis, the floatability of anthracite coal after heating oxidation should decrease due to the increased surface ravines and oxygen-containing groups.

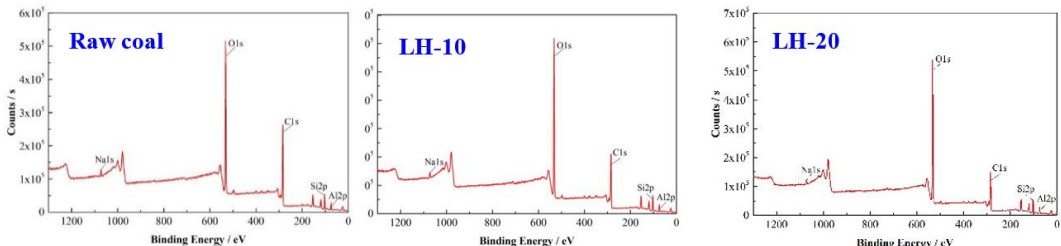

**Figure 7.** XPS wide energy spectrums of coal before and after heating oxidation at 200 °C.

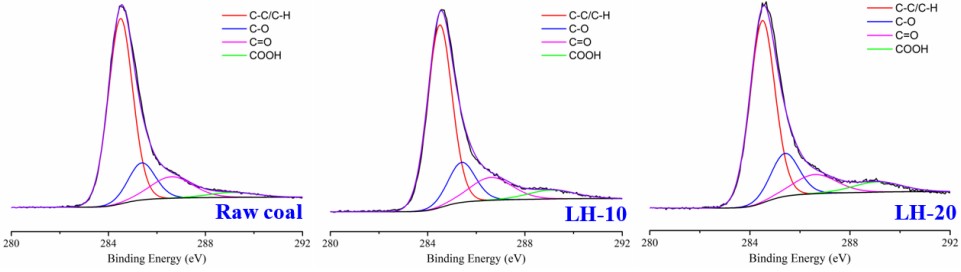

**Figure 8.** Peak fitting of the XPS C1s peak.

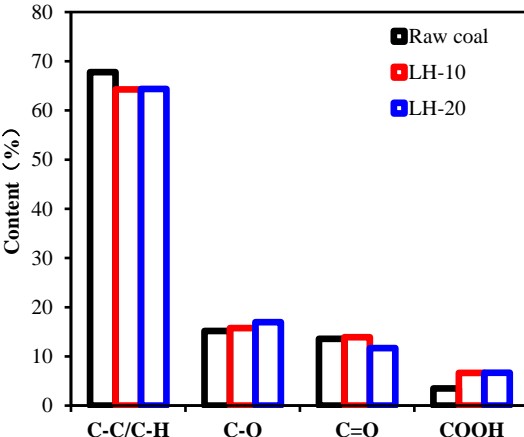

**Figure 9.** Functional groups of coal surface before and after heating oxidation at 200 °C from peak fitting of the XPS C1s peak.

*3.2. Effect of Heating Oxidation at 200 °C on the Floatability of Anthracite Coal*

The effect of oxidation on the induction time of anthracite coal is shown in Figure 10. The induction time significantly increased from 200 ms to 1200 ms and to 2000 ms, illustrating that coal floatability decreased after heating. On the one hand, hydrogen bonding could be easily formed between water molecules and the oxygen-containing groups on coal surfaces. On the other hand, the surface ravines were filled by water, preventing the bubble-particle attachment. These two factors led to film thinning and made a rupture between coal particle and bubble more difficult. More time is needed to complete the attachment process.

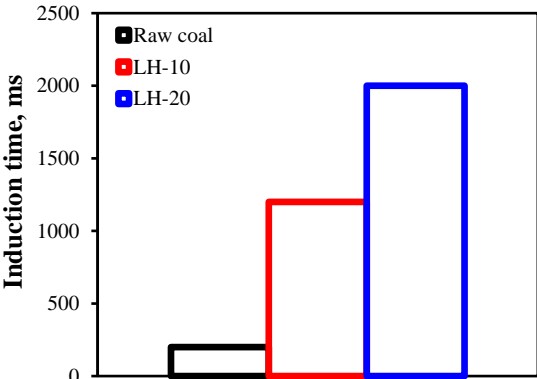

**Figure 10.** Induction time before and after heating oxidation at 200 °C.

The bubble-particle wrap angle before and after heating oxidation at 200 °C is shown in Figure 11. Given that the hydrodynamics and bubble-particle diameter were kept constant in all tests, the attachment angle kinetics was only determined by the competition between bubble-particle attachment and detachment efficiency, which is more suitable to describe the floatability compared with the induction time. The Bubble-particle warp angle had been used by Xia et al. [22,27] to confirm the floatability of low-ranked coal in the presence of different collectors. Raw coal clearly exhibited the fastest kinetics of the bubble-particle wrap angle compared with LH-10 and LH-20. The maximum attachment angle of raw coal was 61.0° while the maximum angle of LH-10 and LH-20 decreased to 40.2° and 30.3°, respectively. The increased oxygen-containing groups make coal hydrophilic, thereby leading to a decreased attachment and an increased detachment efficiency. These results directly illustrated that the floatability decreased after heating. In the study of Çınar [18], the floatability had been investigated by low-temperature heat treatment. After heating it was found that the floatability, hydrophobicity, and separation efficiency of hydrophilic coal changed dramatically. Chang [12]

reported that coal flotation performance decreased with an increase in the degree of surface oxidation, which is consistent with our results.

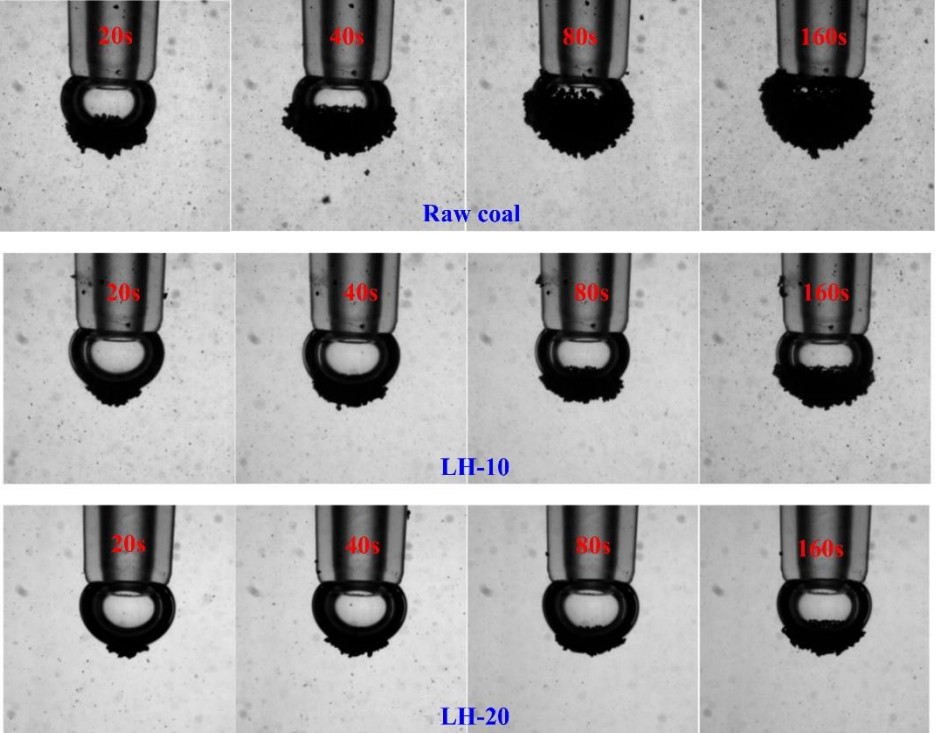

**Figure 11.** Bubble-particle wrap angle kinetics before and after heating oxidation at 200 °C.

## 4. Conclusions

The effect of heating oxidation at 200 °C on the surface/interface characters and floatability of anthracite coal were comprehensively studied in this paper. After heating oxidation, both surface ravines and oxygen-containing groups were observed to increase. The degradation of hydroxyl on anthracite could be neglected during the heating, while the oxidation of hydrocarbon chains dominated the balance of hydrophobicity and hydrophilicity on coal surface. The induction time significantly increased from 200 ms to 1200 ms and 2000 ms after oxidation intervals of 10 h and 20 h by heating at 200 °C, respectively. Furthermore, raw coal exhibited the fastest kinetics of bubble-particle attachment and the largest attachment angle, directly illustrating that the floatability decreased after heating oxidation at 200 °C. Overall, it was observed that the effect of heating oxidation at 200 °C on the floatability of anthracite coal was similar to that of high-temperature oxidation.

**Author Contributions:** Y.X. and G.R. conceived and designed the experiments; M.X. and D.W., performed the experiments; X.G. and G.R. analyzed the data; Y.X. and G.R. wrote the paper.

**Funding:** National Key R&D Program of China (No.: 2018YFC0604702), Jiangsu Province Science Fund for Distinguished Young Scholars (No.: BK20180032), China Postdoctoral Science Foundation funded project (2018M642369), Young Elite Scientists Sponsorship Program by CAST (2018QNRC001), and the National Nature Science Foundation of China (Grant No.: 51774286, 51574236).

**Acknowledgments:** This research was supported by National Key R&D Program of China (No.: 2018YFC0604702), Jiangsu Province Science Fund for Distinguished Young Scholars (No.: BK20180032), China Postdoctoral Science Foundation funded project (2018M642369), Young Elite Scientists Sponsorship Program by CAST (2018QNRC001), and the National Nature Science Foundation of China (Grant No.: 51774286, 51574236), for which the authors express their appreciation.

**Conflicts of Interest:** The authors declare no conflict of interest.

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
