# Peer review of "Effect of Heating Oxidation on the Surface/Interface Properties and Floatability of Anthracite Coal"

_processes, doi:10.3390/pr7060345_

Round 1
Reviewer 1 Report
Accept in present form.
Author Response
Point 1: Accept in present form.
Response 1: Thanks for your review on our paper.

Reviewer 2 Report
The paper reports on a set of laboratory experiments for the study of “Effect of low-temperature oxidation by heating on the surface/interface characters and floatability of anthracite coal”.
It is an interesting work, studies the effect of low-temperature oxidation by heating on the surface/interface characters and floatability of anthracite coal. The paper is well written and the results clearly presented. This paper has interesting and could be accepted with minor changes.
I have listed some comments below.
Why did you not perform some flotation laboratory tests? (for raw coal, after oxidation intervals of 10 h and 20 h).
Line 29
coal; ; Induction must be coal; Induction
Line 56
Carboxyl , must be carboxyl,
Line 80
(EDS) . must be (EDS).
Line 125
“Given that the contact angle of raw coal was < 90°”
Is the contact angle of fresh coal greater or lower than 90º?
Xia W. and Xie G. (2014) verified that the contact angle of fresh coal is about 112°.
Author Response
Point 1: Why did you not perform some flotation laboratory tests? (for raw coal, after oxidation intervals of 10 h and 20 h).
Response 1: Thanks a lot for your helpful comments on our paper. In this study, induction time and bubble-particle wrap angle (i.e. bubble-particle attachment angle) are used to verify the floatability of coal particles before and after heating oxidation.
The bubble-particle wrap angle (i.e. bubble-particle attachment angle) was first introduced by hu [Chu et al., Minerals Engineering, 2014] to indicate the mass of particles attached to bubble surfaces. It was defined as the average coverage angle of the particles onto the bubble surface. The wrap angle test simulates the particle adhesion behavior on a single bubble in a flotation system. The bubble-particle wrap angle is, of course, directly proportional to particle floatability. Larger bubble-particle wrap angle and attachment kinetics corresponded to better floatability.
The test result of induction time and bubble-particle wrap angle can reliably character the floatability of raw coal, after heating oxidation intervals of 10 h and 20 h.
Many researchers perform flotation laboratory tests on oxidation coal. It was found that flotation recovery was well correlated with the induction time and the maximum BPAA [Xing et al., Powder Technology, 2019 and Xia et al. Fuel, 2019]. Flotation experiment is studied from macroscopic level, while induction time and wrap angle are studied from microscopic level on flotation processes.
Point 2: Line 29 coal; ; Induction must be coal; Induction
Response 2: Thanks a lot for your valuable suggestion on our paper. We modify our paper accordingly.
Point 3: Line 56 Carboxyl , must be carboxyl,
Response 3: Thanks a lot for your valuable suggestion on our paper. We modify our paper accordingly.
Point 4: Line 80 (EDS) . must be (EDS).
Response 4: Thanks a lot for your valuable suggestion on our paper. We modify our paper accordingly.
Point 5: Line 125 “Given that the contact angle of raw coal was < 90°” Is the contact angle of fresh coal greater or lower than 90º? Xia W. and Xie G. (2014) verified that the contact angle of fresh coal is about 112°.
Response 5: Thanks a lot for your valuable comments on our paper. The contact angle is different among different coal samples.
In our paper, the anthracite sample was obtained from Xuehu Plant, China, and the ash content was 21.30%. It should be noted that the ash (mineral) is usually hydrophilic. In Xia W. and Xie G.’s paper, an ultra-low-ash anthracite coal with ash content of 1.55% was used. The different of ash content between these two coal samples result in the difference in the hydrophobicity (contact angle). The contact angle of raw coal was approximate 80-90°, showing a relative high hydrophobicity. The relevant result in Xia W. and Xie G.’s paper has been discussed in this paper.
We have corrected our manuscript in detail according to the comments. The revisions are marked in red. Thank you very much for your contributions to our paper.

Reviewer 3 Report
Using the term "low-temperature" in the title, and thought the manuscript is a bit confusing. Low or high are subjective terms, and something which is "low" for the authors may be "high" for some readers. Therefore, the title and aim of the work should be revised. I understand from the text that by "low" the authors mean 200 degree which is a temperature less than what is already been investigated (500 degrees). One cannot find anything relevant to this temperature in the Methodology section.
The other weak point of this manuscript is the discussion section. There are no citation or comparison with the published work in several pages of discussion (pages 4-9). Consequently, their Conclusion is not valid since it is based on a weak discussion.
Author Response
Point 1: Using the term "low-temperature" in the title, and thought the manuscript is a bit confusing. Low or high are subjective terms, and something which is "low" for the authors may be "high" for some readers. Therefore, the title and aim of the work should be revised.
I understand from the text that by "low" the authors mean 200 degree which is a temperature less than what is already been investigated (500 degrees). One cannot find anything relevant to this temperature in the Methodology section.
Response 1: Thanks for your helpful comments on our paper. We do fully agree that it is different for different application on the low or high temperature. In this paper, the temperature of heating oxidation was constant, so we can’t use the “low” which may cause confusing. We modify the title and aim of the work.
The temperature of heating oxidation is 200 °C in this study, and we modify relevant content in Methodology section as below:
“2.1. Materials and heating oxidation process
The anthracite sample was obtained from Xuehu Plant, China. The raw samples were crushed to small particles by hammer mill, whose particle size was smaller than 0.5 mm. The ash content of anthracite was 21.30%. A muffle furnace was used for heating the coal sample. The interval of heating were 10 h and 20 h at 200 °C, whose process was the heating oxidation, respectively.”
Point 2: The other weak point of this manuscript is the discussion section. There are no citation or comparison with the published work in several pages of discussion (pages 4-9). Consequently, their Conclusion is not valid since it is based on a weak discussion.
Response 2: Thanks for your helpful comments on our paper. We add citation and comparison in discussion in our revised paper, accordingly.
We have corrected our manuscript in detail according to the comments. The revisions are marked in red. Thank you very much for your contributions to our paper.

Round 2
Reviewer 3 Report
OK